# Is Antibiotic Prophylaxis Necessary before Dental Implant Procedures in Patients with Orthopaedic Prostheses? A Systematic Review

**DOI:** 10.3390/antibiotics11010093

**Published:** 2022-01-12

**Authors:** Angel-Orión Salgado-Peralvo, Juan-Francisco Peña-Cardelles, Naresh Kewalramani, Alvaro Garcia-Sanchez, María-Victoria Mateos-Moreno, Eugenio Velasco-Ortega, Iván Ortiz-García, Álvaro Jiménez-Guerra, Dániel Végh, Ignacio Pedrinaci, Loreto Monsalve-Guil

**Affiliations:** 1Department of Stomatology, University of Seville, 41009 Seville, Spain; orionsalgado@hotmail.com (A.-O.S.-P.); evelasco@us.es (E.V.-O.); ivanortizgarcia1000@hotmail.com (I.O.-G.); alopajanosas@hotmail.com (Á.J.-G.); lomonsalve@hotmail.es (L.M.-G.); 2Science Committee for Antibiotic Research of Spanish Society of Implants (SEI—Sociedad Española de Implantes), 28020 Madrid, Spain; k93.naresh@gmail.com; 3Department of Basic Health Sciences, Rey Juan Carlos University, 28922 Madrid, Spain; 4Fellow Oral and Maxillofacial Surgery Department and Prosthodontics Department, School of Dental Medicine, University of Connecticut Health, Farmington, CT 06030, USA; 5Department of Nursery and Stomatology, Rey Juan Carlos University, 28922 Madrid, Spain; 6Department of Oral Health and Diagnostic Sciences, School of Dental Medicine, University of Connecticut Health, Farmington, CT 06030, USA; ags.odon@gmail.com; 7Department of Clinical Specialties, Faculty of Dentistry, Complutense University of Madrid, 28040 Madrid, Spain; mateosmoreno80@hotmail.com; 8Department of Prosthodontics, Semmelweis University, 1085 Budapest, Hungary; vegh.daniel.official@gmail.com; 9Department of Dentistry and Oral Health, Division of Oral Surgery and Orthodontics, Medical University of Graz, 8010 Graz, Austria; 10Section of Graduate Periodontology, Faculty of Dentistry, Complutense University of Madrid, 28040 Madrid, Spain; ignpedri@ucm.es; 11Department of Restorative Dentistry and Biomaterials Science, Harvard School of Dental Medicine, Harvard University, Boston, MA 02115, USA

**Keywords:** antibiotic prophylaxis, antibiotics, joint replacement, prosthetic joint infection, oral implantology, dental implants

## Abstract

As the population ages, more and more patients with orthopaedic prostheses (OPs) require dental implant treatment. Surveys of dentists and orthopaedic surgeons show that prophylactic antibiotics (PAs) are routinely prescribed with a very high frequency in patients with OPs who are about to undergo dental procedures. The present study aims to determine the need to prescribe prophylactic antibiotic therapy in patients with OPs treated with dental implants to promote their responsible use and reduce the risk of antimicrobial resistance. An electronic search of the MEDLINE database (via PubMed), Web of Science, LILACS, Google Scholar, and OpenGrey was carried out. The criteria used were those described by the PRISMA^®^ Statement. No study investigated the need to prescribe PAs in patients with OPs, so four studies were included on the risk of infections of OPs after dental treatments with varying degrees of invasiveness. There is no evidence to suggest a relationship between dental implant surgeries and an increased risk of OP infection; therefore, PAs in these patients are not justified. However, the recommended doses of PAs in dental implant procedures in healthy patients are the same as those recommended to avoid infections of OPs.

## 1. Introduction

Today, life expectancy is higher than ever before due to the declining mortality rate of young people in developing countries [1], while “developed” countries are experiencing a large decline in mortality at older ages, with an average life expectancy of 91 years for women and 86 years for men [2]. This increases the number of people in need of orthopaedic prostheses (OPs) of any kind [3]. In 2006, around 800,000 joint replacements (hip and knee, as well as elbow, wrist, ankle, metacarpophalangeal, and interphalangeal joints) were placed in the USA [4] and it is estimated that by 2030 these numbers will increase to 4 million for hip and knee replacements alone [5]. Globally, the age group with the highest prevalence of tooth loss is 79-year-olds. Future figures may be even higher, as between 1990 and 2015, there was a 40% increase in the number of people with oral conditions such as untreated caries and/or tooth loss [6]. For these reasons, it is expected that more and more professionals will be faced with patients with OPs requiring the replacement of missing teeth with dental implants.

Infection of orthopaedic prostheses (OPIs) occurs in 0.3 to 2% of patients with OPs [7], requiring additional orthopaedic surgery, as well as the use of antibiotics for a long period [5]. Depending on the timing of their occurrence concerning orthopaedic surgery, OPIs are classified as “early”, “delayed”, or “late”. The first two occur before three months, and between 3 and 24 months post-surgery, respectively, and are related to orthopaedic surgery. On the other hand, late OPIs often result from the late growth of bacteria accidentally inoculated during bleeding procedures or from a distant septic focus [8,9] (“hematogenous orthopaedic prosthetic infections” (HOPIs)). It is estimated that around 10% of these infections are caused by bacteria present in the oral cavity [3]. However, the frequency, duration, and intensity of bacteraemia will influence the cumulative risk [8]. In this regard, classic animal model studies have shown that high bacterial counts (>1000 colony-forming units (CFU)/mL) are required for HOPIs to occur, which is often a consequence of systemic sepsis [10,11].

Various surveys have shown that 63.4% [12] to 71.5% [13] of orthopaedic surgeons consider the prescription of prophylactic antibiotics (PAs) to be necessary indefinitely in patients with hip prostheses who are going to undergo dental treatment. A survey conducted in Canada in 2014 revealed that around 70.7% of dentists routinely prescribe Pas and 21.7% consider it essential for the rest of the patient’s life [12]. Despite that, currently, antibiotic prophylaxis for patients with prosthetic joints who are undergoing dental treatment is not routinely recommended in several countries, such as Australia [14], the United Kingdom [15], Canada [16], and New Zealand [17]. Namely, a recent survey of professionals dedicated to oral implantology studied for the first time their prescription habits for PAs in patients with hip prostheses, showing that 74.3% prescribe them, as they consider these patients to be at risk [18]. 

Considering the current context, these data should be considered where antimicrobial resistance causes around 33,000 deaths each year in the European Union [19]. The associated healthcare costs and lost productivity are estimated to be 1.5 billion euros per year [20]. It is a naturally occurring phenomenon, and this process is being accelerated by the inappropriate and indiscriminate use of antibiotics in humans, food-producing animals, and the environment. Immediate changes in the way antibiotics are prescribed and used are urgently needed. Even if new methods are developed, resistance will continue to pose a severe threat if current prescribing patterns are not modified [21]. In this sense, a recent meta-analysis revealed that the average prescribed dose of PAs in implant surgery is approximately 5 times higher than the one recommended to healthy patients without anatomical constraints [22]. Furthermore, there is only clear scientific proof on the recommended PA dose in the clinical situation mentioned above [23] and in bone augmentation with the implant placement done in one or two phases [21], that is, 2–3 g of amoxicillin an hour before the intervention [21,23], while in allergic patients, 500 mg of azithromycin, 1 h before surgery, has recently been suggested. [24]. Regarding the remaining clinical situations, the type of antibiotic prescribed and its posology is up to the professional, who in many cases tends to over-prescribe them. 

Therefore, given the data described above, it is considered necessary to carry out a literature review to determine the need for PAs in patients with OPs who are going to be treated with dental implants to promote their responsible use.

## 2. Materials and Methods

The criteria used are the ones described in the PRISMA^®^ (Preferred Reporting Items for Systematic Reviews and Meta-Analyses) Declaration [25].

### 2.1. Focused Question

The main objective was to answer the following “PICO” (P = patient/problem/population; I = intervention; C = comparison; O = outcome) question (Table 1).

In patients with OPs who are about to undergo an implant procedure, does the prescription of PAs decrease the risk of infection of OPs versus not taking them?

### 2.2. Eligibility Criteria

Before proceeding, inclusion and exclusion criteria were defined and applied to the resulting articles.

#### 2.2.1. Inclusion Criteria

The included studies comprised (a) human studies; (b) articles published in English or Spanish (c); meta-analysis; (d) systematic reviews; (e) randomized clinical trials (RCTs); (f) clinical trials; (g) clinical studies; (h) comparative studies; (i) multicentre studies; (j) observational studies; and (k) grey literature.

#### 2.2.2. Exclusion Criteria

The exclusion criteria determined the exclusion of the following: (a) experimental laboratory studies; (b) animal studies; (c) studies whose main topic was not the prescription of PAs before the dental procedure in patients with joint replacements; (d) duplicated articles; (e) books or chapters of books; (f) letters to the Editor; (g) commentaries; (h) literature reviews; and (i) surveys.

### 2.3. Information Sources and Search Strategy

A comprehensive search of the literature was conducted in the following databases: MEDLINE (via PubMed), Web of Science, Google Scholar, and LILACS. A search for unpublished studies (grey literature) was conducted on the OpenGrey database. Moreover, we examined the bibliographic references of the selected articles for publications that did not appear in the initial search and might be of interest. 

The search was performed by two independent researchers (A.-O.S.-P. and J.-F.P.-C.). The search was temporarily restricted from 2 February 2011 to 2 February 2021, and was later updated on 16 February 2021.

MeSH (Medical Subject Headings) terms, keywords and other free terms were used with Boolean operators (OR, AND) to combine searches: (hip prosthesis OR hip replacement OR prosthesis joint OR joint replacement OR knee prosthesis OR knee replacement OR ankle prosthesis OR ankle replacement OR shoulder prosthesis OR shoulder replacement OR elbow prosthesis OR elbow replacement) AND (dental implant OR dental implants OR dental implantology OR oral implantology OR oral surgery OR dental OR dental treatment) AND (antibiotics OR preventive antibiotics OR antibiotic prophylaxis OR clindamycin OR amoxicillin OR erythromycin OR azithromycin OR metronidazole). The same keywords were used for all search platforms following the syntax rules for each database.

### 2.4. Study Records

Two researchers (A.-O.S.-P. and J.-F.P.-C.) independently compared the results to ensure completeness and removed duplicates. Then, the full title and abstracts of the remaining papers were screened individually. Finally, full-text articles included in this systematic review were selected according to the criteria described above. Disagreements over eligible studies to be included were discussed with a third reviewer (N.K.), and a consensus was reached. The reference list of the included studies was also reviewed for possible inclusion.

### 2.5. Risk of Bias

Data collection was conducted using a predetermined table to assess the resulting articles. Two independent reviewers (J.-F.P.-C. and A.G.-S.) evaluated the methodological quality of eligible studies following the Joanna Briggs Institute Checklist for Systematic Reviews and Research Syntheses [26], which incorporates 11 domains. The studies were classified as low-quality assessment studies (0–6) or as high-quality assessment studies (7–11).

## 3. Results

### 3.1. Study Selection

The search strategy resulted in 106 results, of which 99 remained after removing the duplicates. Then, two independent researchers (A.-O.S.-P. and J.-F.P.-C.) reviewed all the titles and abstracts and excluded 86 that were outside the scope of this review. Thus, we obtained 13 potential references. After reading the full text of those 13 papers, it was found that none answered the PICO question as they did not investigate the need to prescribe PAs in implant treatments to reduce the risk of OPIs. Six articles were included that investigated the risk of bacteraemia secondary to dental procedures with varying degrees of invasiveness, so extrapolations were made to establish clear guidelines on the need to administer PAs in implant procedures in these patients. The same situation occurred after the search in Google Scholar and the analysis of the references of the selected articles. Three articles were included in this way so that a total of four papers were analysed [5,7,27,28] (Figure 1).

### 3.2. Study Characteristics

Most of these studies focused on the appropriateness of the prescription of PAs in hip and/or knee prostheses [5,27] and joints in general. The main findings were described as follows. 

The American Dental Association (ADA) and the American Academy of Orthopaedic Surgeons (AAOS) have collaborated on the development of Clinical Practice Guidelines (CPGs) in patients with OPs. The first collaboration was in 2013 [5], where it was determined that it was not necessary to routinely prescribe PAs for dental procedures in patients with hip or knee replacements because, although PAs have been shown to reduce bacteria associated with dental procedures, no evidence links these bacteraemias to HOPIs. Nevertheless, the authors expressed that this decision should be left to the discretion of the professional and the patient after weighing the benefits/risks. This workgroup concluded that PAs would be “rare” or “perhaps appropriate” for non-invasive treatments, meaning those that do not require gingival/mucosa manipulation/perforation, as well as for invasive treatments in healthy patients. Among severely immunocompromised patients, patients with non-controlled diabetes (with levels of glycosylated haemoglobin (HbA1c) ≥ 8 or blood glucose level ≥ 200 mg/dL) and/or patients with a history of OPI, the prescription of PAs would be considered “appropriate”. Patients with severely immunosuppressed states can be classified into the following groups, according to the Center for Disease Control and Prevention (CDC) guidelines [29]: (1) patients with stage III HIV/AIDS, i.e., patients with a CD4 T-lymphocyte count < 200 or opportunistic infections; (2) patients on chemotherapy with fever (absolute neutrophil count (ANC) < 2000) (39 °C) or severe neutropenia (ANC < 500) with or without fever; (3) patients with rheumatoid arthritis (RA) on treatment with disease-modifying biologic agents, including tumour necrosis factor-alpha (TNF-α) or prednisone > 10 mg/day (4) patients who have received a solid organ transplant and are on immunosuppressants; (5) patients with hereditary immunosuppressive diseases; and (6) patients with a bone marrow transplant from the pre-transplant period until the end of immunosuppressive treatment (usually about 36 months after surgery). 

Subsequently, the ADA [27] (2015) carried out a CPG with their workgroup following previous guidelines, not systematically recommending the PAs. Despite this, they suggest assessing the administration in diabetic or immunocompromised patients, including in the immunocompromised group those with antibiotic resistance or under treatment with systemic steroids/immunosuppressive drugs, the presence of some type of cancer, and/or with a history of chronic renal disease. The odd ratios (OR) related to the mentioned systemic alterations varied between 1.8 and 2.2 [30], although the magnitude of these values lacks clinical relevance [27]. Thus, this recommendation should be treated carefully. Also, they suggest assessing PAs on patients with a history of complications associated with joint replacement surgeries that will go through an invasive dental procedure and having an interview with the patient and a consultation with the orthopaedic surgeon. If it is favourable, the latter should recommend the adequate antibiotic prescription and, ideally, issue the pharmacological recipe. 

Subsequently, the Canadian Agency for Drugs and Technologies in Health [28] (CADTH) (2016) conducted a CPG updating a previous consensus document [31] where they concluded that, although there were some cases of HOPIs after dental procedures, most of these infections were not caused by microorganisms present at the oral level and there was insufficient evidence to claim that taking PAs before a dental procedure could prevent HOPIs. Therefore, they do not recommend its prescription in patients with total joint replacements or with orthopaedic pins, plates, or screws. 

In 2017, the Dutch Orthopaedic and Dental Societies [7] conducted a systematic review, concluding that there was no convincing evidence in the literature to justify PAs to avoid HOPIs. Human studies have not confirmed an increased risk of haematogenous infection in joint prostheses during the first two years after placement. Nonetheless, they did observe a higher susceptibility in those that were between two and five years old. Furthermore, the bleeding cannot be considered an isolated factor of bacteraemia, but rather that the gingival/mucosa (including chewing) manipulations cause a match between positive and negative pressures in the capillaries that could favour the diffusion of bacteria into the bloodstream [32]. Thus, positive capillary pressure could avoid the phenomenon.

### 3.3. Risk of Bias within Studies

Risk of bias and study quality analyses were performed independently by two review authors (J.-F.P.-C. and A.G.-S.). Using the predetermined 11 domains for the methodological quality assessment according to the JBI Prevalence Critical Appraisal Tool [26], we determined that all the papers [5,7,27,28] have a high-quality evaluation (7–11). Table 2 shows a more detailed description of the articles included. 

## 4. Discussion

Throughout the past few years, the recommendations regarding the need to prescribe PAs in dental procedures to avoid OPIs have suffered diverse variations. In this regard, the first guidelines recommended prescribing PAs during the first two years after orthopaedic surgery [32]. Some years later, they suggested carrying it out for the rest of the individual’s life [33] and, at present, some countries, such as Australia [14], the United Kingdom [15], Canada [16] and New Zealand [17], advise against its routine prescription. Undoubtedly, OPI is an important complication for the patient and with a high cost for the healthcare system, so the tendency, of oral and/or maxillofacial surgeons and orthopaedic surgeons to prescribe PAs before invasive dental procedures is understandable [8]. Despite this, their consistent use is currently not justified, as there is no evidence that bacteraemias following dental procedures are directly related to HOPIs [5,30,34]. Some authors found HOPI rates after total knee arthroplasty of 22.1%, out of which one-third occurred in immunocompromised patients, 4.2% were related to dental procedures, and 7.2% to cutaneous infections. In the remaining cases, they did not find a causal factor [35]. Thus, their incidence is very low (2.9%; 6/224) and usually related to dental abscesses [36], and often it is hard to prove the relationship between a HOPI to the carried out dental treatment. In this regard, the expected benefits do not exceed the possible adverse effects that could develop due to the consumption of antibiotics [37]. In this sense, it has been determined that the risk of suffering an OPI after a dental procedure without PAs is 0.00023 (0.00012–0.00034), while the OR associated with the PAs prescription is 0.7 [9].

Kao et al. [38] carried out a cohort study of 57,066 patients with knee or hip prostheses undergoing invasive dental treatments. They were compared to patients that had not undergone dental treatment using a ratio of 1:1, describing OPIs rates of 0.57% and 0.61%, respectively. The multivariable analysis of the COX regression did not find any relationship between the invasive dental procedures and OPIs. On the other hand, OPIs happened in 0.20% of patients of the subcohort who were prescribed with PAs and in 0.18% who were not prescribed PAs, without significant differences. 

In implant surgery, there are three moments when bacteraemia can be caused: during (1) the injection of local anaesthesia, (2) the rise of mucoperiosteal flaps, and (3) the placement of the dental implant [39]. Regarding the anaesthetic technique, the infiltrative methods, which are the ones used in oral implantology, produce a significantly lower proportion of positive blood cultures than the modified and conventional intraligamentous techniques (16% vs. 50% vs. 97%, respectively)—the level of basal bacteraemia is 8% [40]. Lockhart [41] suggested that using local anaesthesia with epinephrine could decrease the transit of bacteria to the bloodstream by reducing the flow. Furthermore, the rate of secondary bacteraemia to implant surgery is very low, which was only described by Piñeiro et al. [39] The authors analysed secondary bacteraemia with the placement of implants in patients that rinsed with chlorhexidine digluconate (CLX) at 0.2% (10 mL for 1 min) (test group) before the anaesthetic injection versus a control group that did not use any type of antimicrobial. The level of bacteraemia in the control group was 2% versus 6.7% in the test group. They only observed differences between patients with positive and negative cultures concerning the intervention time (92.5 ± 24.7 min vs. 64.8 ± 20.2 min, respectively), but not so for the history of periodontal disease, the level of oral health, or the number of placed implants. These surgeries were carried out with general anaesthesia, as well as a local one, which has been related to a higher risk of bacteraemia compared to just administering local anaesthesia at 30 s (89% vs. 53%; OR = 5.04), 15 min (64% vs. 24%; OR = 5.37), and 60 min (21% vs. 4%; OR = 6.5) [42].

On the other hand, Watters et al. [5] identified bacteria associated with a higher risk of OPIs, such as *Staphylococcus* spp. (31.5%), specifically, *S. aureus* (26.0%) and *S. epidermidis* (6.5%), Gram-positives species (9.0%), and *Streptococcus* spp. (6.5%). The isolated species after secondary bacteraemia of implant placement were *Streptococcus viridans* and *Neisseria* [39], which are not directly related to OPs complications. Also, certain oral hygiene procedures carried out daily by patients have a risk of associated bacteraemia that is not inconsiderable, with higher proportions of OPI-causing bacteria than in invasive procedures related to oral implantology [5]. 

Noori et al. [43] (2019) established for the first time some recommendations in patients treated with foot and ankle surgeries, including total ankle arthroplasty. The bacteria implicated in post-surgery infections in hip and/or knee joint prosthesis infections are similar to those responsible for foot and ankle surgeries, so they determined that the recommendations should be the same. 

If it is decided to prescribe PAs, the first-choice antibiotic would be 2 g of amoxicillin, 30–60 min before the procedure and, to those allergic to penicillin, 500 mg of azithromycin or clarithromycin [7,39]. Currently, there are only recommendations based on evidence regarding the placement of unitary implants in ordinary conditions [23] and bone regeneration with the placement of implants in one or two phases [21]. In both procedures, the recommended guideline is 2 or 3 g of amoxicillin, one hour before the intervention [21,23] and, in allergic patients, 500 mg of azithromycin one hour before surgery [24], which would be equivalent to what is recommended to prevent OPIs. 

The current guidelines recommend assessing PAs in non-controlled diabetic patients, those immunocompromised, and/or in patients with a history of OPI. For a long time, it has been presumed that immunocompromised patients have a higher risk of HOPIs. Nonetheless, it has not been proved after dental procedures, since they develop comparable bacteraemia to the ones developed by healthy individuals, and with the latter, there is no evidence that there is a higher risk [7]. On the other hand, the European Association of Endodontics [44] recommends prescribing PAs in patients with OPs during the three months following orthopaedic joint surgery. Dental implants entail an elective surgery, so it does seem prudent to postpone the intervention during this time. Also, dental implant surgery is not indicated for the described systemic states until the disease has been controlled. The only scenario where the treatment could be assessed in these patients would be in those with AR being treated with biological agents modifying the disease, and despite this, different authors associated it with a higher risk of early placement failure and peri-implantitis [45,46].

The relationship between the periodontal state, or the level of hygiene, and the level of bacteraemia was not established [9,39]. Nonetheless, any implant procedure must be done in an oral cavity without any pathology. Another preventive measure is using perioperative antiseptics, such as CLX. Although ADA/AAOS was not able to conclude that rinsing with different topical antimicrobials before a dental procedure prevents HOPIs [5], further studies found that CLX rinses reduce the incidence of secondary bacteraemia in dental extractions by 12% [47,48,49], possibly due to a reduction of the quantity of inoculated bacteria. This is a simple preventive measure without evidence of adverse reactions. Thus, its use is recommendable despite its low efficacy [47]. Furthermore, as a chemo-preventive measure against the accumulation of biofilm, its use is recommended in the immediate post-surgery period, a time when oral hygiene procedures could experience difficulties [50].

Given that the placement of dental implants currently involves the prescription of PAs in healthy patients, future lines of research should focus on establishing the incidence of infections following implant procedures in those patients with OPs. It would also be interesting to study topical antiseptics, such as CLX, to reduce secondary bacteraemia in implant procedures.

### Strengths and Limitations

This systematic review presents several strengths, such as a previous record of protocol, free search in the literature (including grey literature), the searching process of studies, data extraction, and the risk analysis bias performed in duplicate, which determined a high overall quality of the included studies.

Nonetheless, with the low number of studies available in the literature, the present systematic review has limitations, so the external validity of the results of this review should be confirmed with future studies.

## 5. Conclusions

No evidence suggests a relationship between dental implant surgery and a higher risk of infection of OPs. Therefore, the prescription of PAs in these patients is not justified. Nonetheless, the recommended PA dose in a dental implant procedure in healthy patients is comparable to the dose recommended to avoid infections in OPs. We should evaluate the prescription of PAs in patients with a history of infections in their OPs in second-stage implant surgeries. Furthermore, it would be wise to avoid surgeries three months after orthopaedic surgery.

## Figures and Tables

**Figure 1 antibiotics-11-00093-f001:**
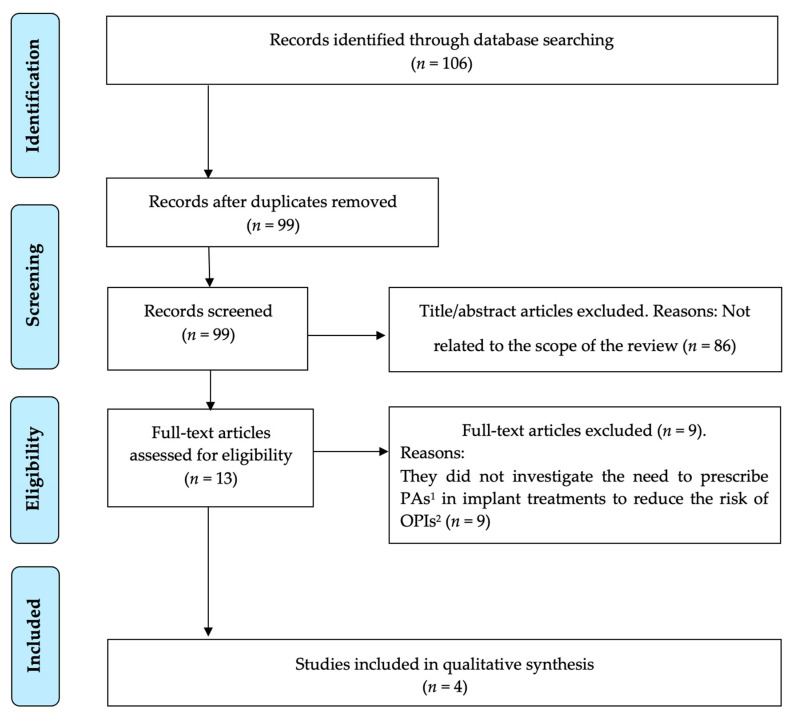
PRISMA^®^ flow diagram of the search processes and results. ^1^ PAs, prophylactic antibiotics; ^2^ OPIs, orthopaedic prostheses infection.

**Table 1 antibiotics-11-00093-t001:** Breakdown of the “PICO” question.

Component	Description
P (problem/population)	Patients with OPs ^1^ that have had a dental implant treatment
I (intervention)	PAs ^2^ on the day of surgery and/or extended postoperatively
C (comparison)	Not prescribing PAsPrescribing a placeboOther antibiotics or antibiotic regimensSame antibiotic with different dosage/duration
O (outcome)	Risk of infection from OPsSafety (for example, benefits for the patient, OPIs ^3^ prevention, resistance to antimicrobials)
PICO question	In patients with OPs who are about to undergo an implant procedure, does the prescription of PAs decrease the risk of infection of OPs versus not taking them?

^1^ OPs, orthopaedic prostheses; ^2^ PAs, prophylactic antibiotics; ^3^ OPIs, orthopaedic prostheses infection.

**Table 2 antibiotics-11-00093-t002:** JBI Critical Appraisal Tool [26] for Systematic Reviews and Research Syntheses.

Questions	Rademacher et al. [7] (2017)	CADTH [28] (2016)	Sollecito et al. [27] (2013)	Watters et al. [5] (2013)
1. Is the review question clearly and explicitly stated?	^  ^	^  ^	^  ^	^  ^
2. Were the inclusion criteria appropriate for the review question?	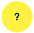	^  ^	^  ^	^  ^
3. Was the search strategy appropriate?	^  ^	^  ^	^  ^	^  ^
4. Were the sources and resources used to search for studies adequate?	^  ^	^  ^	^  ^	^  ^
5. Were the criteria for appraising studies appropriate?	^  ^	^  ^	^  ^	^  ^
6. Was critical appraisal conducted by two or more reviewers independently?	^  ^	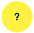	^  ^	^  ^
7. Were there methods to minimize errors in data extraction?	^  ^	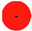	^  ^	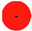
8. Were the methods used to combine studies appropriate?	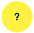	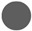	^  ^	^  ^
9. Was the likelihood of publication bias assessed?	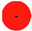	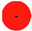	^  ^	^  ^
10. Were policy and/or practice recommendations supported by the reported data?	^  ^	^  ^	^  ^	^  ^
11. Were the specific directives for new research appropriate?	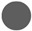	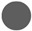	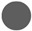	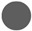


—Yes; 
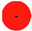
—No; 
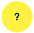
—Unclear; 
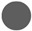
—Not applicable.

## Data Availability

Data are available in a publicly accessible repository.

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
