# Peer review of "Is Antibiotic Prophylaxis Necessary before Dental Implant Procedures in Patients with Orthopaedic Prostheses? A Systematic Review"

_antibiotics, 2022, doi:10.3390/antibiotics11010093_

Round 1

Reviewer 1 Report

The paper is well written and interesting.  The problem of antibiotic prophylaxis is urgent and a lot of dentists abuse of antibiotic. The study is well organized and the problem has been deeply addressed. On line 98 "that is, 2–3 g of amoxicillin an hour before the intervention" I suggest to add tha in case of allergies different  antibiotics must be used. No problems with plagiarism (up to 12%). I suggest just an English mother-language revision.

Author Response

Reviewer #1:

Dear reviewer,

Thank you for taking the time to review our manuscript and helping us improve the quality of our work. After analysing your comments, we proceed to answer the one by one:

The paper is well written and interesting.  The problem of antibiotic prophylaxis is urgent and a lot of dentist’s abuse of antibiotic. The study is well organized, and the problem has been deeply addressed. On line 98 "that is, 2–3 g of amoxicillin an hour before the intervention" I suggest adding that in case of allergies different antibiotics must be used. No problems with plagiarism (up to 12%).

The recommended dose in allergic patients (azithromycin 500 mg, 1 h before surgery) was added (lines 100 and 327).

I suggest just an English mother-language revision.

English was checked again.

Reviewer 2 Report

This paper addresses an important topic which often leads to confusion in clinical practice. The paper is well written and the flow of logic throughout the paper is strong. This is likely to be a highly cited article when it appears.

There are two minor additions required for the paper and two minor errors to be corrected.

Minor additions

On line 80 of the introduction please give a year and a location for the study (12) which is cited in support of the following statement “ Around 70.7% of 80 dentists routinely prescribe PAs and 21.7% consider it essential for the rest of the patient's life”. This is extremely important context since recommendations around the use of PAs arose before the concept of antimicrobial stewardship. As well, many countries now have explicit recommendations against the use of PAs for patients with orthopaedic prostheses.

In the paper please include a list of countries and professional associations and academies that currently have recommendations against the routine use of PAs for patients with orthopaedic prostheses. This will better help readers of the paper to appreciate the international perspective on the research question. It would also add to the current discussion which focuses only on Europe and North America.

Minor errors

In the abstract on line 46 change “therefore, Pas in these patients is not justified” to “therefore, PAs in these patients is not justified” by capitalising the letter A.

The indentation for reference 46 is incorrect.

Author Response

Reviewer #2:

Dear reviewer,

Thank you for taking the time to review our manuscript and helping us improve the quality of our work. After analysing your comments, we proceed to answer the one by one:

This paper addresses an important topic which often leads to confusion in clinical practice. The paper is well written and the flow of logic throughout the paper is strong. This is likely to be a highly cited article when it appears.

There are two minor additions required for the paper and two minor errors to be corrected.

Minor additions

On line 80 of the introduction please give a year and a location for the study (12) which is cited in support of the following statement “Around 70.7% of 80 dentists routinely prescribe PAs and 21.7% consider it essential for the rest of the patient's life”. This is extremely important context since recommendations around the use of PAs arose before the concept of antimicrobial stewardship. As well, many countries now have explicit recommendations against the use of PAs for patients with orthopaedic prostheses.

The location (Canada) and year (2014) were added (lines 77-78).

In the paper please include a list of countries and professional associations and academies that currently have recommendations against the routine use of PAs for patients with orthopaedic prostheses. This will better help readers of the paper to appreciate the international perspective on the research question. It would also add to the current discussion which focuses only on Europe and North America.

After the evaluation of the reviewer's suggestion, several countries that do not routinely recommend PAs for patients with orthopaedic prostheses before dental treatments, like Canada, Australia, United Kingdom and New Zealand, among others (lines 79 to 82, and 264 to 266) were added.

Minor errors

In the abstract on line 46 change “therefore, Pas in these patients is not justified” to “therefore, PAs in these patients is not justified” by capitalizing the letter A.

It was modified (line 43).

The indentation for reference 46 is incorrect.

It was modified.

Reviewer 3 Report

First of all i want to thank the authors for their submission, lthink several adjustments must be made before the article can be evaluated for submission, Ihave attached my comments below.

  1. The article should be submitted to a professional proofing service as many phrases are quite hard to follow.
  2. I think the title should be changed given the characteristics of the paper and of the included studies.
  3. Moreover, the topic of the article is totally different from the one exemplified in the title as the included articles do not focus on the topic (the application of antibiotics in patients with orthopaedic prostheses).
  4. The major flaw of the article is the low amount of articles included in the review, and the characteristics of these articles, which are all reviews of their own. I would advise the authors to repeat the review process and the article selection broadening their PICO question.

Author Response

Reviewer #3:

First, I want to thank the authors for their submission, I think several adjustments must be made before the article can be evaluated for submission, I have attached my comments below.

1. The article should be submitted to a professional proofing service as many phrases are quite hard to follow.

English was checked again.

2. I think the title should be changed given the characteristics of the paper and of the included studies.

3. Moreover, the topic of the article is totally different from the one exemplified in the title as the included articles do not focus on the topic (the application of antibiotics in patients with orthopaedic prostheses).

Suggestions 2 and 3 are answered jointly as they are along the same lines. Thus, the authors appreciate the comments of reviewer #3, however, millions of implants are placed worldwide every year. The fact that the literature available to date has not established recommendations on whether or not antibiotic prophylaxis should be prescribed in patients with orthopaedic prostheses who are to undergo implant procedures does not mean that it is not necessary to do so. Antimicrobial resistance due to indiscriminate and inappropriate use of antimicrobials is currently a threat, as many diseases that were under control will once again become life-threatening. Therefore, the authors consider it essential to establish guidelines to serve as a starting point for future studies. Nevertheless, the title was modified.

4. The major flaw of the article is the low number of articles included in the review, and the characteristics of these articles, which are all reviews of their own. I would advise the authors to repeat the review process and the article selection broadening their PICO question.

The authors regret to disagree with the suggestion made since, as questions 2 and 3 were answered, the fact that no specific studies are answering the PICO question does not mean that there is no need to answer it, as it is precisely an issue that needs to be shed light on. Furthermore, the PICO question itself was formulated broadly and, despite not having found articles that met the established inclusion criteria and specifically answered the PICO question, it has been possible to draw clear conclusions from the discussion. There are two pieces of information that we would like to highlight in this respect:

  1. It has been determined that the risk of suffering an OPI after a dental procedure without PAs is 0.00023 (0.00012 - 0.00034), while the OR associated with the PAs prescription is 0.7 (Moreira et al. 2020) (lines 279 - 280).
  2. Another study also showed that the bacteria associated with an increased risk of orthopaedic prosthesis infections are Staphylococcus spp. species, which differ from bacteria isolated after bacteraemia secondary to implant surgery (lines 309 - 313).

Round 2

Reviewer 3 Report

I want to thank the authors for their answers, useful for publication of the review.

But I would ask to underline strongly the limitations of the number of studies included and the proper recomendations for the clinicians,

Author Response

Dear reviewer 3,

Thank you for taking the time to review our manuscript and helping us improve the quality of our work. After analysing your comments, we proceed to answer them:

The authors appreciate the comments of reviewer #3. Millions of implants are placed worldwide every year. It is fascinating that to this day with all the research that has been done in the field of dentistry, there is still no recommendation in the available literature as to whether or not antibiotic prophylaxis should be prescribed in patients with orthopaedic prostheses who are to undergo implant procedures. The fact that there are currently no recommendations does not mean that there is no need for consensus. Antimicrobial resistance due to indiscriminate and inappropriate use of antimicrobials is currently a global problem, as many diseases that were thought to be under control will once again become fatal. Therefore, the authors consider it essential to establish guidelines to serve as a starting point for future studies and we intend to motivate professionals to create new future lines of research to develop a consensus on this field of research. 

Regards,

The authors